# The Prevalence of Daily Sedentary Time in South Asian Adults: A Systematic Review

**DOI:** 10.3390/ijerph18179275

**Published:** 2021-09-02

**Authors:** Kamalesh C. Dey, Julia K. Zakrzewski-Fruer, Lindsey R. Smith, Rebecca L. Jones, Daniel P. Bailey

**Affiliations:** 1Institute for Sport and Physical Activity Research, School of Sport Science and Physical Activity, University of Bedfordshire, Bedford MK41 9EA, UK; kamalesh.dey@study.beds.ac.uk (K.C.D.); lindsey.smith@beds.ac.uk (L.R.S.); rebecca.jones@beds.ac.uk (R.L.J.); daniel.bailey@brunel.ac.uk (D.P.B.); 2Division of Sport, Health and Exercise Sciences, Department of Life Sciences, Brunel University London, Uxbridge UB8 3PH, UK; 3Sedentary Behaviour, Health and Disease Research Group, Brunel University London, Uxbridge UB8 3PH, UK

**Keywords:** sedentary behaviour, sitting, cardiometabolic disease, cardiovascular disease, South Asians

## Abstract

This study aimed to systematically review total daily sedentary time in South Asian adults. Seven electronic databases were searched, identifying relevant articles published in peer-reviewed journals between March 1990 and March 2021. The study was designed in accordance with PRISMA guidelines. Prospective or cross-sectional design studies reporting total daily sedentary time in South Asian adults (aged ≥18 years), reported in English, were included. Study quality and risk of bias were assessed, and the weighted mean total daily sedentary time was calculated. Fourteen full texts were included in this systematic review from studies that were conducted in Bangladesh, India, Norway, Singapore, and the United Kingdom. Pooled sedentary time across all studies was 424 ± 8 min/day. Sedentary time was measured using self-report questionnaires in seven studies, with a weighted mean daily sedentary time of 416 ± 19 min/day. Eight studies used accelerometers and inclinometers with a weighted mean sedentary time of 527 ± 11 min/day. South Asian adults spend a large proportion of their time being sedentary, especially when recorded using objective measures (~9 h/day). These findings suggest that South Asians are an important target population for public health efforts to reduced sedentary time, and researchers and practitioners should seek to standardise and carefully consider the tools used when measuring sedentary time in this population.

## 1. Introduction

South Asians are referred to as “Indo-Asians” and are considered the natives of first or subsequent generation migrants originating from the Indian subcontinents (i.e., Bangladesh, India, Pakistan, Nepal, and Sri Lanka) [1]. South Asians are the largest growing population, increasing in population by 1.1% annually and representing one-fifth of the current global population [2]. At present, around 463 million adults live with diabetes worldwide (approximately 90% of these cases being Type 2 diabetes), with numbers expected to rise to 700 million by 2045 [3]. South Asians are suggested to be up to six times more likely to develop Type 2 diabetes than other ethnicities [4]. In the UK, the prevalence of Type 2 diabetes in South Asians is approximately 20%, which is almost five-fold higher than in Caucasian adults [5]. Likewise, the prevalence of Type 2 diabetes is three times higher in migrant South Asians living in the USA and Canada than other ethnic groups [6,7,8]. South Asians also have significantly higher (three- to five-fold) cardiovascular disease (CVD) incidence, myocardial infarction, and mortality risk than other ethnicities [4,9,10,11], and the standardised mortality rate for ischaemic heart disease (IHD) is higher in UK-based South Asians than Caucasians and African–Caribbean adults [12]. Possible reasons for this increased risk in South Asians include migration and demographic transitions to a more Westernised lifestyle, dietary intake (e.g., high consumption of saturated fat), visceral adiposity, low physical activity levels, and higher sedentary time [5,10,13,14,15,16].

Sedentary behaviour is any waking behaviour characterised by an energy expenditure of ≤1.5 metabolic equivalents (METs) while in a reclining, lying, or sitting posture [17]. Engaging in a high amount of sedentary time significantly increases the risk of CVD, Type 2 diabetes, and all-cause mortality in the general population [18,19,20,21,22]. A systematic review of 22 studies found that greater sedentary time was associated with an increased risk of diabetes and cardiometabolic risk markers in South Asian adults [16]. This may be of concern if South Asians engage in high volumes of sedentary behaviour. However, the systematic review by Ahmad et al. [16] did not report the sedentary time of the studies included, meaning that it is unclear how sedentary South Asian adults are at a population level. A number of studies, however, have reported daily sedentary time in South Asians. For instance, approximately 64% of Bangladeshi and Pakistani females living in the UK spent an average of 8.8 h per day being sedentary when measured using objective methods [23]. The World Health Organization (WHO) STEPS survey reported that the prevalence of self-report daily sedentary time among Bangladeshi, Indian, Pakistani, Nepali, and Sri-Lankan adults was 175, 309, 223, 202, and 216 min, respectively [24], which could be considered relatively low. Objective methods for the measurement of sedentary time include accelerometers and inclinometers. Accelerometers are worn on a fixed point of the body (typically wrist, hip or thigh) and quantify acceleration resulting from bodily motion [25]. Inclinometers are devices that measure posture and are worn on the thigh to distinguish between sitting, standing, and lying [25]. A previous systematic review examining physical activity and sedentary time profiles in South Asians reported that objectively measured sedentary time ranged from 482 to 587 min/day [26]. However, this review included only accelerometer-derived data. Accelerometers cannot detect posture and, therefore, may misclassify standing as sedentary time, thus leading to overestimations [27]. Furthermore, Mahmood et al. [26] pooled data from South Asians and African Surinamese adults from more than 10 countries (i.e., Bangladesh, Indian, Pakistan, Nepal, Sri Lanka, Maldives, Bhutan, Afghanistan, the Netherlands, and African countries) rather than focusing exclusively on South Asians, who may have distinct sedentary time profiles. The review by Mahmood et al. [26] also did not quantitatively synthesise sedentary time across the reported studies, so it is not clear what the mean sedentary time of South Asians is across the literature.

To contribute a more comprehensive understanding of the prevalence of daily sedentary time in the South Asian population, this systematic review aimed to examine the total daily amount of time spent sedentary quantified via both subjective and objective measures in South Asian adults.

## 2. Materials and Methods

### 2.1. Design

This review was designed according to the Preferred Reporting Items for Systematic Reviews and Meta-Analysis (PRISMA) guidelines [28], and the study protocol was registered with PROSPERO (CRD42017065778). Ethical approval was obtained from the Institute for Sport and Physical Activity Research Ethics Committee, University of Bedfordshire, before starting the study (approval number: 2018ISPAR011).

### 2.2. Study Search and Selection

Seven electronic databases (PubMed, Web of Science, Biomed Central, Medline, Scopus, SPORTDiscus, and Cumulative Index of Nursing and Allied Health Literature [CINAHL]) were systematically searched for research published between March 1990 and March 2021. The following Boolean operators were used: (“sitting” OR “sedentar*”) AND (“South Asian*” OR “Bangladesh*” OR “Indian*” OR “Pakistan*” OR “Sri Lanka*” OR “Nepal*”). The search was limited to peer-reviewed journal articles published in English. Only cross-sectional or prospective study design articles were considered for inclusion, with review articles and conference abstracts being excluded. Studies were eligible for inclusion if they reported data on a South Asian population (either born in their resident country or migrated from the Indian subcontinent); included adults (aged ≥18 years); measured total daily sedentary time using self-report or objective methods; the subjects were defined as healthy at baseline; provided a clear definition of sedentary time (e.g., not defined as “physical inactivity”); and reported total sedentary time in minutes or hours per day. Authors were contacted by email to provide South Asian sub-group data if data in the article were pooled with other ethnic groups; these studies were not included if the requested data were not provided. Studies were excluded if they were a review or intervention study, did not clearly define the ethnicity of the participants, or did not conform to ethical standards.

Following the removal of duplicates, a three-phase search strategy was subsequently employed by two reviewers (K.C.D. and D.P.B.). Firstly, the eligibility of the study titles was screened, and, secondly, abstracts were screened against the eligibility criteria. Thirdly, full articles were retrieved and assessed against the inclusion/exclusion criteria. The reference lists of relevant original and review articles were screened to identify any additional relevant studies.

### 2.3. Data Extraction and Analysis

Data extraction was conducted by K.C.D. The following data were extracted: author, year of publication, study design, sample characteristics (age, ethnicity, and sex), country of study, South Asian sample size, method of sedentary behaviour measurement, and mean sedentary time per day. A previous systematic review in the general population found that >6 h/day of daily sedentary time was the threshold at which the risk of CVD mortality increased [21]. To provide a practical health implication for the present review of South Asian adults who experience increased CVD risk, a threshold for a daily sedentary time of >360 min/day (6 h/day), indicating “high” sedentary time, was applied.

### 2.4. Risk of Bias and Study Appraisal

Two reviewers (K.C.D. and D.P.B.) independently assessed the quality and risk of bias according to a standardised set of predefined criteria (see Table 1) [29,30]. The criteria consisted of eight items, with each item carrying a score weight of one: (1) sufficient description of the source population, (2) sufficient description of the sampling frame, recruitment methods, period of recruitment, and place of recruitment (setting and geographic location), (3) participation rate at baseline ≥80%, or if the non-response was not selective, (4) sufficient description of the study sample, (5) measurement method for sedentary time, (6) total sedentary time, (7) presentation of point estimates and measures of variability: standard deviation (SD), confidential interval (CI) or standard error, and (8) no selective reporting of results. Each quality criterion was rated as positive (scored as 1) and negative (scored as 0). The final score was agreed upon mutually between two reviewers (K.C.D. and D.P.B.). A study was considered “high-quality” and “low-quality” if the quality assessment score was ≥7 and <7, respectively.

### 2.5. Synthesis of Results

The available literature was systematically reviewed to calculate the total amount of sedentary time (min/day) in South Asians using a narrative and analytical approach. A narrative synthesis was conducted in this review because of the different types of sedentary behaviour measurement methods (e.g., self-reported questionnaire, accelerometers, and inclinometers) rather than a meta-analysis. The weighted mean was calculated using the following formula [31]: Weighted mean (Xm) = (∑WiX)/(∑Wi), where Wi is the weighting value (sample size/total sample size), and X is the sedentary time. The weighted mean for total daily sedentary time was calculated separately for studies presenting self-reported and objective measures of sedentary time. Sedentary time was expressed in min/day. Similarly, the weighted SD was calculated using the formula [31]: Weighted SD (X_SD_) = (∑WiX)/(∑Wi), where Wi is the weighting value (sample size/total sample size), and X is the SD for sedentary time. All data were analysed in Appendix A and expressed as weighted mean ± weighted SD unless stated otherwise.

## 3. Results

### 3.1. Study Characteristics

The search identified a total of 2501 studies, with 2297 studies remaining after removing duplicates. The study title screening led to the exclusion of 2128 studies, with an abstract review excluding 118 studies. Fourteen published studies met the criteria for inclusion after a full-text screening. Therefore, 14 studies were included in this systematic review (Figure 1). All included studies were published between 2011 and 2020 and originated from five different countries (Bangladesh [*n* = 1], India [*n* = 2], Norway [*n* = 1], Singapore [*n* = 4], and the UK [*n* = 6]; see Table 2); 3 studies used a prospective cohort study design [32,33,34], and the other 11 studies used a cross-sectional design [23,35,36,37,38,39,40,41,42,43,44].

Of the UK-based studies, the participants included were migrant Bangladeshis [23,39], Pakistanis [23,39,40], Indians [39,40] and UK-born South Asians [23,39]. Three UK-based studies recruited migrant South Asians but did not specify their nationality [34,36,37]. The four Singaporean studies recruited migrant Indians living in Singapore but did not specify the participants’ birth location [32,33,38,44]. Two studies conducted in India recruited native Indians living in urban and rural areas in India [41,43]. The Norway-based study recruited participants born either in Pakistan or Norway [35], whilst the study conducted in Bangladesh recruited native Bangladeshis living in rural and urban areas [42] (see Table 2).

The sample sizes for the South Asian ethnic groups in the studies ranged widely, from 11 to 6447 South Asian participants (see Table 2). The participants’ age range was 18 to 89 years, apart from one study that used a cut-off of 17 years [43]. As the average age was 41 ± 10 years and only a small number of Indian-based studies were available, this study was included in the results. Eight studies included both males and females [33,34,36,38,40,42,43,44], two studies included only males [35,39], with the remaining four studies including only female participants [23,32,37,41].

### 3.2. Study Quality

The study quality rating for each individual included study is shown Appendix A. The quality scores of the individual studies included in this review ranged from 5 to 8. Eleven of the studies were considered high quality [23,35,36,37,38,39,40,41,42,43,44] and three of the studies were considered low quality [32,33,34].

### 3.3. Sedentary Time Measurement Methods

Total sedentary time was measured using objective measures (i.e., accelerometers and inclinometers) in eight studies (accelerometers: [23,35,37,38,39,41,42]; inclinometers: [36]; self-report questionnaires were used in seven studies: [32,33,34,38,40,43,44]). One study used both a questionnaire and accelerometers [38]. All data was reported as minutes or hours/day (see Table 2).

### 3.4. Total Sedentary Time in South Asian Adults

The pooled data from all studies showed that the total weighted mean sedentary time was 424 ± 8 min/day in South Asian adults. This weighted mean value is higher than the cut-off for “high” sedentary time (360 min/day) used in this review [21]. In total, 11 of the 14 studies reported daily sedentary times that were above this threshold.

Seven studies reported total mean sedentary time ranging between 303 ± 158 and 658 ± 170 min/day using self-report questionnaires [32,33,34,38,40,43,44] (see Table 2). The weighted mean for the self-reported total sedentary time was 416 ± 19 min/day. Based on eight studies using objective methods, total mean sedentary time ranged between 516 ± 134 and 615 ± 534 min/day [23,35,36,37,38,39,41,42] (see Table 2). The weighted mean for the objective measurement methods was 527 ± 11 min/day of sedentary time.

In relation to sedentary time in male and female South Asians, based on self-reported measures, the average sedentary time was similar in Indian males (475 ± 165 min/day) and females (474 ± 161 min/day) in the study by Sullivan et al. [43], although Padmapriya et al. [32] reported lower sedentary time in migrant Indian females (431 ± 193 min/day). Based on objective measures, there was also no clear difference in sedentary time between male and female South Asians. In males, sedentary time ranged from 516 ± 96 to 551 ± 95 min/day [35,39], and, in females, it ranged from 519 ± 87 to 532 ± 102 min/day [23,37,41].

Of the 14 included studies, only 3 studies recruited South Asians who were living in a South Asian country (i.e., Bangladesh and India). The sedentary time of these samples ranged from 474 ± 161 to 551 ± 83 min/day [41,42,43]. The sedentary time of migrant South Asians, according to the other 11 studies, ranged from 303 ± 158 to 658 ± 170 min/day [23,32,33,34,35,36,37,38,39,40,44].

## 4. Discussion

This is the first study, to the authors’ knowledge, to systematically review total daily sedentary time based on both self-report and objective methods in South Asian adults. The review’s main findings are that South Asian adults engage in a mean daily sedentary time of approximately 7 h (424 ± 8 min), although sedentary time varies widely between self-reported and objective measures. The daily sedentary time of South Asians in the present review is lower than the 482 to 587 min/day reported by Mahmood et al. [26]. This discrepancy may be due to Mahmood et al. [26] reporting total mean sedentary time based on accelerometer data only (i.e., ActiGraph and ActiHeart), whereas the current review reports total sedentary time in South Asians based on both objective and subjective measurement methods. However, based on the weighted sedentary time of the studies that used objective methods in the present review, daily sedentary time was 527 min/day, which is similar to the range reported in the systematic review by Mahmood et al. [26].

Sedentary time was relatively higher when measured using objective methods (527 ± 11 min/day) compared with self-report methods (416 ± 19 min/day). Total daily sedentary time was 111 min per day higher according to objective methods. This is not surprising, given that many sitting activities in individuals’ daily lives may be omitted within self-report measures, including eating, driving, phoning, listening to music, writing, and personal care [45]. Other sedentary activities, including typing, playing videos games, and sending text messages, might have also been missed when estimating total sedentary time using self-report measures [46]. Self-reported measures have other limitations, including low levels of data validity and reporting biases, leading to the underestimation of levels of sedentary behaviour due to the lack of conscious processing with sedentary behaviour, which might limit the ability of an individual to recall their sitting time accurately [25,47]. Therefore, overall sedentary time may be underestimated by 2.2 to 3.4 h/day [48]. Accelerometers and inclinometers, on the other hand, can measure activities continuously during the monitoring period and, therefore, capture all sedentary activities (e.g., sitting) that may not be recollected or reported when answering questionnaires [48]. The findings of the present study extend those of a meta-analysis that found self-report methods underestimated sedentary time by 105 min/day compared with device measures in general population adults, but not specifically South Asian adults [49]. However, some monitoring devices (i.e., accelerometers) have limited functional abilities to detect posture and may therefore misclassify standing as sedentary time [27]. The activPAL inclinometer accurately detects posture and may thus be considered the gold standard method for measuring sitting time. In the present review, only one study measured sedentary time using activPAL devices and found that South Asian adults engaged in high amounts of sitting (516 min/day) [36]. According to objective measures in this review, which may provide a more accurate estimation of sedentary time than self-reported measures, South Asian adults spend approximately 9 h each day being sedentary. This could contribute to their increased risk of Type 2 diabetes, CVD, and mortality [18,20,21,22]. Future research is needed to confirm this hypothesis using objective methods of sedentary time that accurately detect this behaviour. The findings of the present review also support those of a systematic review that demonstrated that sedentary time varies widely across studies conducted within the same country and across different European countries [50]. This highlights the need for the standardisation of methodology across studies that seek to understand the sedentary behaviour profiles of different population groups.

Based on self-reported measures, the reported average daily sedentary time within the included studies in this review was similar in Indian males and Indian females, although lower sedentary time was reported in migrant Indian females [32,43]. Similarly, based on objective measures, the average daily sedentary time for the studies in this review was similar among male and female South Asians [23,35,37,39,41]. However, this review was not able to draw any firm conclusions regarding any potential differences in sedentary time between male and female South Asians due to the absence of sedentary time reported separately for each sex, i.e., only three male [35,39,43] and five female samples [23,32,37,41,43] were available for review. There also did not appear to be any consistent evidence regarding any differences in sedentary time in migrants compared with native South Asian populations. However, drawing a firm conclusion in this respect is difficult due to the small number of studies that have investigated native South Asian participants [41,42,43]. It is possible that factors such as culture, religion and the environment could affect the sedentary time of migrant versus native South Asians, as has been suggested in relation to physical activity levels [51]. It is also possible that these factors could mediate the association between sedentary time and CVD, but there is limited research available exploring this issue in South Asian populations [16]. According to this systematic review, there does not appear to be any clear indication as to a difference in sedentary time between male and female South Asians or between migrant and native South Asians. Thus, future studies should include direct comparisons between males and females and migrant and native populations, in addition to investigating how these factors could affect associations with CVD risk. This would help to identify potential population groups that may benefit more from targeted interventions.

The pooled total sedentary time from all studies in South Asians in this review was 424 ± 8 min/day, which is higher than the threshold of 360 min/day identified by Patterson et al. [21], at which the risk of CVD mortality increases significantly. Within the current review, 11 out of 14 studies reported total mean sedentary times that ranged between 402 and 658 min/day [23,32,34,35,36,37,38,39,41,42,43], which are all above this threshold and therefore provide consistent evidence of the potentially increased CVD risk in South Asian adults. The remaining three studies reported mean sedentary time between 303 and 345 min/day in migrant South Asians living in the UK and Singapore [33,40,44]. The reason for the lower sedentary time in these three studies might be because of the measurement methods employed, i.e., self-reported measures, which can underestimate sedentary time significantly [48]. However, other studies using self-report methods found higher daily sedentary time [32,34,38,43], meaning the measurement method may not be the only reason for the disparate findings. A combination of factors such as country of study, migration status and measurement methods could affect sedentary time estimates. Nonetheless, the literature reviewed here suggests sedentary time could be high in South Asians, to the level that may increase their risk of CVD mortality. Potential reasons for a high sedentary time in South Asians could be related to a lack of knowledge around the risks of engaging in high sedentary time, cultural norms and a lack of awareness or understanding of what constitutes sedentary behaviour [39,52].

### Limitations of the Study

The current review incorporates some studies that have a small sample size, meaning that certain groups may have been under-represented (e.g., limited scope for exploring sedentary time based on income, education, and employment status for South Asians living in different countries), resulting in limited generalisability of the results for some socio-demographic sub-groups. Of particular importance is that a direct comparison between male and female and native and migrant South Asians was not possible across the studies due to the lack of studies that have reported separate sitting times for these groups. Furthermore, it is crucial to recognise that South Asians are a diverse ethnic group originating from five different South Asian nations, with substantial differences based on language, culture, religion, diet, and lifestyle. Consequently, there needs to be some caution in generalising the results of these included studies to all South Asians. Although the literature search for this review was conducted for studies published up until March 2021, which spanned the COVID-19 pandemic, all of the studies included were conducted before the pandemic. Thus, potential changes in sedentary time due to COVID-19 did not influence the findings. Future research should, thus, evaluate the effects of COVID-19 on sedentary time in South Asian adults, which could be higher due to national restrictions such as home confinement and social distancing, as has been reported in other population groups [53].

## 5. Conclusions

This systematic review concludes that the prevalence of daily sedentary time in South Asian adults is influenced by the measurement methods used, with objective measures resulting in 111 min/day higher sedentary time than self-report measures. Regardless of the measurement method used, South Asians should be an important target population for public health strategies focused on reducing sedentary time. There is currently insufficient evidence regarding any differences in sedentary time between male and female or migrant and native South Asians. These findings suggest that researchers and practitioners should seek to standardise and carefully consider the tools used when measuring sedentary time in this population to appropriately inform public health guidelines.

## Figures and Tables

**Figure 1 ijerph-18-09275-f001:**
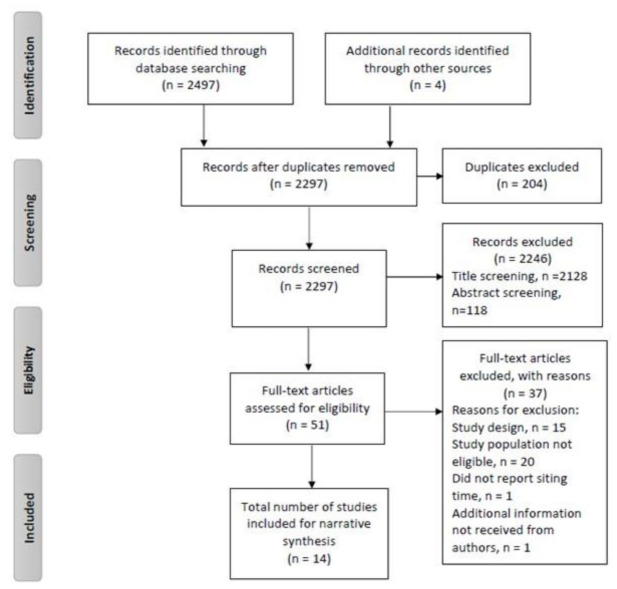
PRISMA flow chart of the study selection process.

**Table 1 ijerph-18-09275-t001:** Methodologic quality assessment of the included studies.

Criteria ^a^	I, V/P	Studies Meeting Criteria n (%)
**Study population and participation (baseline)**		
1. Sufficient description of source population ^b^.	I	14/14 (100)
2. Sufficient description of the sampling frame, recruitment methods, period of recruitment, and place of recruitment (setting and geographic location) ^b^.	I	10/14 (71)
3. The participation rate at baseline ≥ 80%, or if the non-response was not selective, show that the baseline study sample does not significantly differ from the population of eligible participants.	V/P	11/14 (79)
4. Sufficient description of the baseline study sample (i.e., individual participants in the study) for key characteristics (number, age, sex, ethnicity, and sedentary time) ^c,d^.	I	12/14 (86)
**Data collection**		
5. Sufficient measurement of sedentary behaviour: completed by objective measures (i.e., accelerometer, inclinometer), and not by self-report questionnaire (self-report = no; no/inadequate information = unknown).	V/P	8/14 (57)
6. Sedentary behaviour (total sitting time) was measured in min/day or hours/day.	V/P	14/14 (100)
**Data analysis**		
7. Presentation of point estimates and measures of variability (standard deviation, confidential interval, or standard error).	I	14/14 (100)
8. No selective reporting of results.	V/P	14/14 (100)

^a^ Criteria were rated as follows: “yes” refers to an informative description of the criterion at issue, which met the quality criterion; “no” refers to an informative description but an insufficient execution or lack of description of the criterion; and “unknown” refers to a vague or incomplete description of the criterion. ^b^ Sufficient = necessary information to be able to repeat the study; ^c^ yes is given only if sufficient information is given on all criteria. ^d^ yes is given only if no selective dropout on key characteristics is reported in the text or tables. I, criterion on informativeness; V/P, criterion on validity/precision.

**Table 2 ijerph-18-09275-t002:** Overview of study characteristics.

Author and Year	Country	Sample Size, Ethnicity, Age and Sex	Study Design	Sedentary Time Measurement Method	Total Mean Daily Sedentary Time (min/day)	Higher or Lower than the Threshold for High Sedentary Time
(360 min/day)
(Patterson et al., 2018) [21]
Andersen et al. (2011) [35]	Norway	150 Pakistanis (either born in Norway or Pakistan)	Cross-sectional study	Accelerometer	516 ± 96	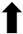
37 ± 7 years	(worn on the right hip)
100% male
Biddle et al. (2019) [36]	United Kingdom	289 South Asians	Cross-sectional study	Inclinometer	516 ± 134	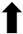
55 ± 11 years	(worn on the midline anterior aspect of the upper thigh)
43% female
Castaneda et al. (2018) [37]	United Kingdom	25 migrant South Asians	Cross-sectional study	Accelerometer	532 ± 102	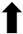
70 ± 8 years	(worn on the hip)
100% female
Chu et al. (2018) [38]	Singapore	11 Indians	Cross-sectional study	Questionnaire and	658 ± 170 (questionnaire)	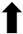
20–65 years	Accelerometer	615 ± 53 (accelerometer)
69% female	(worn on the waist)
Curry and Thomson (2014) [23]	United Kingdom	140 South Asians	Cross-sectional study	Accelerometer	519 ± 87, born in the UK.	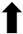
(33 born in the UK, 49 born in Bangladesh, and 58 born in Pakistan)	(worn on the waist)	523 ± 91, born in Bangladesh.
46 ± 14 years	539 ± 71, born in Pakistan.
100% female
Emadian and Thompson (2017) [39]	United Kingdom	54 South Asians (either born in the UK or migrants from Bangladesh, India, and Pakistan)	Cross-sectional study	Accelerometer	551 ± 95	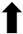
44 ± 9 years	(worn on the right hip)
100% male
Gill et al. (2011) [40]	United Kingdom	1228 South Asians	Cross-sectional study	Questionnaire	345 ± 17	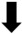
(Indian and Pakistani)
49 ± 10 years
523 males and 705 females
Mathews et al. (2013) [41]	India	47 Indians	Cross-sectional study	Accelerometer	519 ± 115	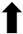
46 ± 3 years	(worn on the right hip)
100% female
Mumu et al. (2017) [42]	Bangladesh	155 Bangladeshis	Cross-sectional study	Accelerometer	551 ± 83	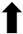
35 ± 9 years	(worn on the waist level above the left hip)
54% females
Padmapriya et al. (2015) [32]	Singapore	209 Indians	Cohort study (prospective)	Questionnaire	431 ± 193	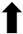
≥18 years
100% female
Sullivan et al. (2011) [43]	India	6447 Indians	Cross-sectional study	Questionnaire	475 ± 165 for male	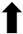
41 ± 0.2	474 ± 161 for female
3768 males and 2679 females
Uijtdewilligen et al. (2017) [33]	Singapore	2385 Indians	Cohort study (prospective)	Questionnaire	303 ± 158	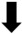
≥21 years
54% male
Vaingankar et al. (2020) [44]	Singapore	366 Indians	Cross-sectional study	Questionnaire	345 ± 194	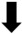
40 ± 14 years
Males and females
Yates et al. (2012) [34]	United Kingdom	97 South Asians	Cohort study (prospective)	Questionnaire	402 ± 468	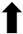
59 ± 10 years
52 males and 45 females

Data presented as mean ± SD; 
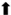
, above the threshold value for high sedentary time; 
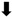
, below the threshold value for high sedentary time.

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
