# Peer review of "The Prevalence of Daily Sedentary Time in South Asian Adults: A Systematic Review"

_ijerph, 2021, doi:10.3390/ijerph18179275_

Round 1
Reviewer 1 Report
As a cardiologist I highly appreciate the topics focusing on cardiovascular risk factors and especially the ones employing the profound analyses as it was in this case. My recommendations are as follows:
- An interesting point in the review was mentioning in methods accelerometer and inclinometer use; I generally think that this could enrich the paper if the authors shortly described both of them in 1-2 sentences, or eventually consider including a photography.
- The mean of sedentary time was very high and reached about 7 hours a day; although authors mentioned the diversity of ethnic groups I suggest to add to discussion section at least a couple of sentences explaining to the reader why in South Asian adults the result is like that (culture, religion, diet, habits etc.) supported by relevant references.
- The defined time in methodology was till March 2021, so it includes also Covid-19 pandemic, please add comment if the studies presented included this time, if no, at least, please, add a comment on the fact that during pandemic the sedentary daily time could even be prolonged, also applying 1-2 references from selected up-to date articles from pandemic time.
Author Response
Reviewer 1
As a cardiologist I highly appreciate the topics focusing on cardiovascular risk factors and especially the ones employing the profound analyses as it was in this case. My recommendations are as follows:
Response: We would like to thank the reviewer for their positive and constructive comments. We have addressed the recommendations as described below. Please note that the line numbers were not visible to some of the authors in the manuscript that had undergone journal formatting as part of the review process. With this in mind, we have included line numbers and page numbers in our responses below in the event that the Reviewers experience similar issues; further, all of our revisions have been tracked using ‘tracked changes’ and should, thus, be clear within the manuscript.
- An interesting point in the review was mentioning in methods accelerometer and inclinometer use; I generally think that this could enrich the paper if the authors shortly described both of them in 1-2 sentences, or eventually consider including a photography.
Response: Thank you for this suggestion. A description of these devices has been provided on lines 69-72 (page 2).
- The mean of sedentary time was very high and reached about 7 hours a day; although authors mentioned the diversity of ethnic groups I suggest to add to discussion section at least a couple of sentences explaining to the reader why in South Asian adults the result is like that (culture, religion, diet, habits etc.) supported by relevant references.
Response: Thank you for this suggestion. Discussion around the potential reasons for high sedentary time has now been included on lines 313-319 (pages 2 and 11).
- The defined time in methodology was till March 2021, so it includes also Covid-19 pandemic, please add comment if the studies presented included this time, if no, at least, please, add a comment on the fact that during pandemic the sedentary daily time could even be prolonged, also applying 1-2 references from selected up-to date articles from pandemic time.
Response: Thank you for noticing this. None of the studies included were conducted during COVID-19. We have added this to the discussion and included a discussion around sedentary time potentially being higher during the pandemic (see lines 331-337; page 11).
Reviewer 2 Report
The prevalence of daily sedentary time in South Asian adults: A systematic review
Fourteen full texts were included in this systematic review from the studies were conducted in Bangladesh, India, Norway, Singapore, and the United Kingdom.
- So you are studying adults of South Asian origin in these countries overseas, not South Asian in South Asian countries.
This may place them at an increased risk of cardiovascular disease mortality.
- You need to compare the risk of CVD in South Asian and other ethnic groups by locations (countries) and observed sedentary time, because you were studying adults of South Asian origin in these countries overseas, and assume sedentary time is associated with CVD risk. How could ethnicity, culture, locations and other factors affect the morbidity and mortality of CVD?
A systematic review of 22 studies found that greater sedentary time was associated with an increased risk of diabetes and cardiometabolic risk markers in South Asian adults [19].
A previous systematic review examining physical activity and sedentary time profiles in South Asians reported that objectively measured sedentary time ranged from 482 to 587 min/day [22].
- I just think only reporting sedentary time of an ethnic group does not give much health implications because research is all about comparison.
- In the intro you have stated that South Asian were at higher risk of having CVD or DM
- What increase risk of CVD?
- -- South Asian ethnicity? Or actually their diet/food source due to culture? (yes/no)
- -- location? (yes/no)
- -- sedentary, with cutoff? (yes/no)
- You can have a structured comparison like South Asian, in Asia, sedentary vs.. Caucasian, overseas, active OR 2x2x2 multiple ways to compare.
- I mean you need to do a review or report a review by linking both CVD and either one of the factors or more than one of them to show that you can arrive at a health implication.
- Why didn’t you criticize reference [19]? They showed association/correlation between sedentary time and disease risk in South Asian but this is not enough because the association may be more stronger in other ethnic groups, unless the situation is worse in South Asian then you can claim that your topic is important.
- Given in the studies that you have reviewed, can you extract more information to make more useful comparisons and adjust you focus a little bit?
The following Boolean operators were used: (“sitting” OR “sedentar*” OR 84 “television”) AND (“South Asian*” OR “Bangladesh*” OR “Indian*” OR “Pakistan*” OR 85 “Sri Lanka*” OR “Nepal*”).
- Nowadays I think it is not only TV, many people work in front of computers like you. How about smartphone and tablets? TV games? "screen-based" devices?
March 1990 and March 2021.
- Pay attention that 1990 we did not have iPhone yet. It is now 2021.
Table 1.
- So this is the point needs explanation, were studies which did not meet the criteria of "data collection" included in synthesis? Why?
Table 2
- I just read the paper of Patterson, " Conclusions: Independent of PA, total sitting and TV viewing time are associated with greater risk for several major chronic disease outcomes. For all-cause and CVD mortality, a threshold of 6-8 h/day of total sitting and 3-4 h/day of TV viewing was identified, above which the risk is increased. "
- Total sitting could be up to 480 min, which could change the arrows of your table 2. Not table 3 continue.
Discussion
Mean sedentary time 211 of South Asians is thus considered high in relation to the threshold applied (360 min/day) 212 that indicates an increasing risk of CVD mortality [17].
- As I have explained above I do not think you can make health implication at this stage of your manuscript.
The evidence thus appears to be consistent in suggesting 222 that South Asian adults engage in high volumes of sedentary behavior.
- I do think when you say high, there must be a comparison group, if South Asian sit for 9 hours a day, how much time would other population spend on sitting?
- Or just our lifestyle changed in the human race?
Self-reported measures 231 have other limitations including low levels of data validity and reporting biases leading 232 to underestimating levels of sedentary behavior
- I see that limitation of self report data become discussion, however, it should be a well known limitation, pls focus on the association between sedentary behavior and health concern in South Asian populations globally in your discussion.
- Another important thing in your discussion following this line is that, actually different studies use different measuring methods, not one study compare both methods, how could you make a conclusion here? Except reference [34], see questionnaire has higher sedentary time.
According to the literature, there does not ap- 258 pear to be any clear indication as to a difference in sedentary time between male and fe- 259 male South Asians, but this should be explored in future studies that include direct com- 260 parisons in the samples studied.
- Yes
- do think about how useful of the results would be when researchers do their study.
Furthermore, it is crucial 286 to recognize that South Asians are a diverse ethnic group originating from five different 287 South Asian nations with strong differences based on language, culture, religion, diet, and 288 lifestyles. Consequently, there needs to be some caution in generalizing the results of these 289 included studies to all South Asians.
- Yes
- Appreciate that researchers could state limitations, most of them just keep saying useless claims without recognizing their limitations which are obvious to peers.
Conclusions
- However, I am conservative to the conclusion that South Asian's sedentary time is high/such behavior is "severe" because I don’t see a clear comparison. If there is high, there should be low.
- Comparing South Asian's sedentary/active behavior and CVD risk with Other's sedentary/active behavior and CVD risk.
- South Asian (yes /no)
- Sedentary (yes /no)
- CVD (yes /no)
Author Response
Reviewer 2
The prevalence of daily sedentary time in South Asian adults: A systematic review
Fourteen full texts were included in this systematic review from the studies were conducted in Bangladesh, India, Norway, Singapore, and the United Kingdom.
Response: We would like to thank the reviewer for their positive and constructive comments. We have addressed the recommendations as described below. Please note that the line numbers were not visible to some of the authors in the manuscript that had undergone journal formatting as part of the review process. With this in mind, we have included line numbers and page numbers in our responses below in the event that the Reviewers experience similar issues; further, all of our revisions have been tracked using ‘tracked changes’ and should, thus, be clear within the manuscript.
- So you are studying adults of South Asian origin in these countries overseas, not South Asian in South Asian countries.This may place them at an increased risk of cardiovascular disease mortality.
- You need to compare the risk of CVD in South Asian and other ethnic groups by locations (countries) and observed sedentary time, because you were studying adults of South Asian origin in these countries overseas, and assume sedentary time is associated with CVD risk. How could ethnicity, culture, locations and other factors affect the morbidity and mortality of CVD?
Response: The aim of this review was to examine the total daily amount of time spent sedentary quantified via both subjective and objective measures in South Asian adults. The search criteria meant that any South Asian population could be included i.e. it was unrestrictive in this sense. This meant that both migrant and native South Asian populations were eligible for inclusion. Of the 14 included studies, only 3 studies recruited native South Asians i.e. South Asians living in a South Asian country, as described on lines 225-229 (please see pages 5 and 9). We have now included a discussion around this on lines 313-319 (page 10), which includes reference to potential reasons as to why sedentary time could differ based on migration status/country of residence.
It was beyond the scope of this review to directly compare the risk of CVD in relation to sedentary time in South Asian versus other ethnic groups. We agree that this would be valuable but it would not currently be possible due to the very limited amount of studies that provide within-study comparisons of the association of sedentary time with CVD in South Asian versus other population groups. This has been discussed on lines 273-277 (page 10).
- A systematic review of 22 studies found that greater sedentary time was associated with an increased risk of diabetes and cardiometabolic risk markers in South Asian adults [19].
- A previous systematic review examining physical activity and sedentary time profiles in South Asians reported that objectively measured sedentary time ranged from 482 to 587 min/day [22].
- I just think only reporting sedentary time of an ethnic group does not give much health implications because research is all about comparison.
Response: The main aim of this review was to examine the total daily amount of time spent sedentary quantified via both subjective and objective measures in South Asian adults. We have shown that sedentary time overall is higher based on objective compared with subjective methods. This has important health implications as researchers and practitioners are better informed in relation to how sedentary time should be measured and interpreted in South Asian populations. Our findings extend those of similar systematic reviews and meta-analyses that (a) found self-report measures underestimated sedentary time compared with device measures in general population adults, and (b) that sedentary time varies widely across studies conducted within the same country and across different countries; this is discussed on lines 246 to 256 and 267 to 271 (page 10). The conclusions have also been edited to bring these more in line with the focus of the review.
We agree that it would be valuable to compare sedentary time and CVD risk outcomes between different ethnic groups, but this was beyond the scope of this review and there is insufficient evidence reporting on within-study comparisons of the association of sedentary time with CVD in South Asian versus other population groups. This has been discussed on lines 286-294 (page 10).
- In the intro you have stated that South Asian were at higher risk of having CVD or DM
What increases risk of CVD?
-- South Asian ethnicity? Or actually their diet/food source due to culture? (yes/no)
-- location? (yes/no)
-- sedentary, with cutoff? (yes/no)
Response: Thank you for identifying that we had not included potential reasons for the higher CVD risk. This has been added on lines 47-50 (page 2).
- You can have a structured comparison like South Asian, in Asia, sedentary vs.. Caucasian, overseas, active OR 2x2x2 multiple ways to compare.
- I mean you need to do a review or report a review by linking both CVD and either one of the factors or more than one of them to show that you can arrive at a health implication.
Response: The authors suggest that it would not be appropriate or possible to conduct such a 2x2x2 analysis, or even a 2x2 analysis in a systematic review in this field. This is due to the lack of studies that have include within-study comparisons of different ethnic groups in terms of associations of sedentary time with health outcomes.
- Why didn’t you criticize reference [19]? They showed association/correlation between sedentary time and disease risk in South Asian but this is not enough because the association may be more stronger in other ethnic groups, unless the situation is worse in South Asian then you can claim that your topic is important.
Response: We have now included criticism of this reference on lines 58-60 (page 2), which relates to the lack of description of sedentary time in the included studies, meaning that it is unclear how sedentary South Asian adults are at a population level.
- Given in the studies that you have reviewed, can you extract more information to make more useful comparisons and adjust you focus a little bit?
Response: In line with other recommendations from this reviewer, we have now added additional results and discussion items to provide a more comprehensive understanding of differences in self-report versus objectively measured sedentary time, and the sedentary time of migrant versus native South Asians.
- The following Boolean operators were used: (“sitting” OR “sedentar*” OR 84 “television”) AND (“South Asian*” OR “Bangladesh*” OR “Indian*” OR “Pakistan*” OR 85 “Sri Lanka*” OR “Nepal*”).
- Nowadays I think it is not only TV, many people work in front of computers like you. How about smartphone and tablets? TV games? "screen-based" devices?
Response: It is correct that we included television in the search terms but not other screen-based behaviours. However, upon reflection, we feel that it is most appropriate to remove ‘television’ and not include other screen-based behaviours from the search criteria as this would not give a reflection of total daily sedentary time, which was the outcome for this review. Thus, we have removed ‘television’ from the Boolean operators list and updated the number of papers generated in the search – this did not affect the papers included in the systematic review.
- March 1990 and March 2021. Pay attention that 1990 we did not have iPhone yet. It is now 2021.
Response: Due to the limited literature in South Asians, we have included a timeline between 1990 and 2021 to enable the inclusion of a sufficient number of published studies to form conclusions. That said, the search did not result in the identification of any studies published before 2010. As stated in line 168 (page 5), we have included studies published between 2011 and 2020. It is thus unlikely technological advances in smartphones will have affected the results.
- Table 1. So this is the point needs explanation, were studies which did not meet the criteria of "data collection" included in synthesis? Why?
Response: Table 1 described the quality assessment of only the studies that were included in the review as specified in the Table title. Therefore, studies that did not meet data collection criteria and thus excluded from the review are not represented in this table.
- Table 2. I just read the paper of Patterson, " Conclusions: Independent of PA, total sitting and TV viewing time are associated with greater risk for several major chronic disease outcomes. For all-cause and CVD mortality, a threshold of 6-8 h/day of total sitting and 3-4 h/day of TV viewing was identified, above which the risk is increased. "
- Total sitting could be up to 480 min, which could change the arrows of your table 2.
Response: We thank the Reviewer for highlighting this - Patterson et al. (2018) reported that a threshold of 6 h/day sitting is associated with an increased risk of CVD mortality and 8 h/day sitting is associated with an increased risk of all-cause mortality. We selected to use the CVD mortality threshold as this better reflects the increased CVD risk of South Asian adults. We have provided greater clarity of this on lines 122-126 (page 3).
- Not table 3 continue.
Response: If we understand the reviewer’s point correctly, we would like to confirm the it is indeed the continuation of Table 2 that goes across more than one page.
Discussion
- Mean sedentary time of South Asians is thus considered high in relation to the threshold applied (360 min/day) that indicates an increasing risk of CVD mortality [17].
- As I have explained above I do not think you can make health implication at this stage of your manuscript.
- The evidence thus appears to be consistent in suggesting that South Asian adults engage in high volumes of sedentary behavior.
- I do think when you say high, there must be a comparison group, if South Asian sit for 9 hours a day, how much time would other population spend on sitting?
Response: We appreciate the reviewer’s reservations in relation to making conclusion regarding the health implications of our findings. We have now modified the discussion so that it is clear that we are, based on our results and previous evidence around thresholds for CVD mortality, suggesting that sedentary time could be high in South Asians, as opposed to making a definitive conclusion on this.
- Or just our lifestyle changed in the human race?
Response: We have now referred to such factors that may explain increases in sedentary time in the discussion on lines 310-313 (page 11).
- Self-reported measures have other limitations including low levels of data validity and reporting biases leading to underestimating levels of sedentary behaviour
- I see that limitation of self report data become discussion, however, it should be a well known limitation, pls focus on the association between sedentary behavior and health concern in South Asian populations globally in your discussion.
Response: As per our responses to your valuable suggestion above, we have now reworked much of the discussion to clarify its focus on comparing self-report to objective measures of sedentary time, with some reference to previous evidence to provide an indication as to whether the sedentary time reported could be considered high and have health implications.
- Another important thing in your discussion following this line is that, actually different studies use different measuring methods, not one study compare both methods, how could you make a conclusion here? Except reference [34], see questionnaire has higher sedentary time.
- According to the literature, there does not appear to be any clear indication as to a difference in sedentary time between male and female South Asians, but this should be explored in future studies that include direct comparisons in the samples studied.
- Do think about how useful of the results would be when researchers do their study.
Response: As suggested, we have provided discussion around methods of measurement differing between studies and the lack of within-study comparisons of different samples and methods on lines 298-308 (pages 10-11). We hope that our interpretations now provide a more useful insight for researchers and practitioners and have provided recommendations in this regard on lines 337-339 (page 11).
- Furthermore, it is crucial to recognize that South Asians are a diverse ethnic group originating from five different South Asian nations with strong differences based on language, culture, religion, diet, and lifestyles. Consequently, there needs to be some caution in generalizing the results of these included studies to all South Asians.
- Appreciate that researchers could state limitations, most of them just keep saying useless claims without recognizing their limitations which are obvious to peers.
Response: We are glad that the reviewer appreciates this important limitation of the literature that we have identified during our review.
Conclusions
- However, I am conservative to the conclusion that South Asian's sedentary time is high/such behavior is "severe" because I don’t see a clear comparison. If there is high, there should be low.
- Comparing South Asian's sedentary/active behavior and CVD risk with Other's sedentary/active behavior and CVD risk.
- South Asian (yes /no)
- Sedentary (yes /no)
- CVD (yes /no)
Response: As per our responses to your valuable suggestion above, we have now reworked much of the discussion to clarify its focus on comparing self-report to objective measures of sedentary time, with some reference to previous evidence to provide an indication as to whether the sedentary time reported could be considered high and have health implications. We have modified the conclusion to bring it in line with the main aim and findings of the review.
Round 2
Reviewer 2 Report
.